# Comparing Academic Performance of Elementary Education Majors in General Education Science Courses

**Ryan S. Nixon**[1]*, **Elizabeth Gibbons Bailey**[2]

**1** Department of Teacher Education, Brigham Young University, Provo, Utah, United States of America,
**2** Department of Biology, Brigham Young University, Provo, Utah, United States of America

* rynixon@byu.edu

## Abstract

It is important for elementary teachers to understand the content they are responsible for teaching their students, known as content knowledge. In the content area of science, elementary teacher preparation programs often expect preservice teachers to develop content knowledge in college science courses completed prior to entering the program. These college science courses are often general education courses, not specifically designed for preservice elementary teachers. General education courses may not be adequately serving preservice elementary teachers. The purpose of this study is to explore the impact of general education science courses on preservice elementary teachers, as compared to other students at the same institution. We collected student grades in six different general education courses across ten years of instruction, resulting in a data set with 195860 grades. These data were analyzed using linear mixed modeling to predict course grades in each of the individual courses. Overall, these findings indicate that elementary education majors in general education courses are receiving grades similar to students in most other majors. Notably, elementary education majors received grades comparable to STEM majors in Biology, while scoring worse than STEM majors in Physical Science. These findings assuage some concerns about the impact of general education courses on elementary education majors and suggest that elementary education programs seeking to provide a specialized science course may want to prioritize a course in physical science.

## Introduction

It is important for elementary teachers to understand the content they are responsible for teaching their students [1–3]. This knowledge, known as content knowledge or subject matter knowledge, includes having the skills (e.g., conventions of effective writing) [4] and understandings (e.g., understanding photosynthesis and transpiration) [5] students are expected to learn. Content knowledge involves the teacher being able to "do the work that they assign their students" [6]. Research has found that teachers' content knowledge impacts teachers' self-efficacy [7], instructional practices [8,9], and students' learning [10,11]. Of course, teachers must be knowledgeable and skillful in areas beyond the content (e.g., instructional

**Data availability statement:** All relevant data are within the manuscript and its Supporting Information files.

**Funding:** The author(s) received no specific funding for this work.

**Competing interests:** The authors have declared that no competing interests exist.

strategies, assessment). However, in this study we focus on content knowledge due to its foundational role [2,9].

In the content area of science, elementary teacher preparation programs often expect preservice teachers to develop content knowledge in college science courses completed prior to entering the program (see Fig 1) [12, 13,14]. As noted by Rice [15], there is an assumption that preservice elementary teachers "entering teacher education [programs] have adequate science subject matter knowledge" from their prior coursework (p. 1078). These college science courses are often general education courses, not specifically designed for preservice elementary teachers [7,16].

General education courses may not be adequately serving preservice elementary teachers. Since they are not designed for future elementary teachers, they often emphasize topics that are not aligned with topics taught in elementary schools [17]. For example, a general education physical science course may spend a significant amount of time on nuclear physics and relativity, topics that go well beyond the scope of the elementary curriculum. This could lead to a sense that these courses are not relevant for preservice elementary teachers [18,19]. As a result, elementary education majors may not engage in these courses and learn the content they need.

Furthermore, studies have found that the number of, or success in, college science courses is not associated with other indicators of teacher content knowledge [20–26]. Nowicki, Sullivan-Watts [27], for example, found that the number of college science courses completed was not associated with the accuracy of science content presented in observed lessons. In a review of the research on teacher knowledge in science, van Driel, Hume [9] noted that college science coursework does not guarantee strong science content knowledge.

The purpose of this study is to explore the impact of general education science courses on preservice elementary teachers, as compared to other students at the same institution. To do this, we draw on data from a large sample of students spanning 10 years of enrollment in six different general education courses. Therefore, our research question is: How do the grades of elementary education majors compare to the grades of other students in general education courses?

## Literature review

Science is an important part of the elementary school curriculum. Educational agencies across the world include science as a part of the elementary level curriculum [26]. This is, in part, because research has repeatedly found that children are natural scientists who are interested in and capable of learning science [28,29,30]. Additionally, learning science supports the development of foundational knowledge and skills that are useful for other domains [31].

Despite its importance, science is often treated as less important than other content areas in the elementary school curriculum. Studies in the US find that less time is allocated for science instruction than subjects such as literacy and mathematics [7,32,33]. Teachers regularly report

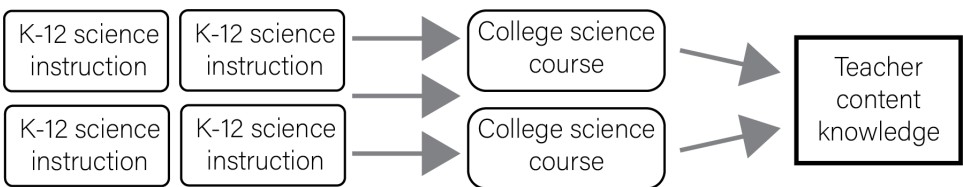

**Fig 1. Model of content knowledge development in many teacher preparation programs.**

having less preparation in science than other content areas and, likely as a result, less confidence in their ability to teach science [7,33]. Furthermore, within the content area of science, elementary teachers tend to be more prepared and more confident with the life sciences than with the physical sciences [33–35]. The mismatch between the importance of children learning science and the status of science in schools necessitates better understanding the preparation of elementary teachers to teach science.

With almost 90% of elementary teachers identifying as female [36–38], it is not surprising that this low prioritization of science and preference for biology over other physical science matches gender differences observed more broadly. It is well documented that female students are more likely to enroll in, and find belonging in, biology college courses as compared to courses in other STEM fields [39–42]. Further, studies have found that males tend to receive higher grades than females in STEM courses, though findings remain mixed [43–46].

An important part of elementary teachers' preparation for teaching science includes college science courses, typically completed prior to coursework specifically related to teaching [7,13,26,30,47]. These courses are frequently general education courses, meaning that they include students from a variety of majors and are part of the university requirements to broaden students' knowledge [7,16].

An ongoing concern with general education courses for students from all majors is students' perception of relevance [18,19,48,49]. Because general education courses are designed for a general audience, it is not immediately obvious how they are related to students' interests or pursuits [50]. Researchers have explored ways to help general education courses be perceived as more relevant by students, including institution-wide strategies such as improving communication about the purposes of general education [51] or restructuring the general education program [52,53]. More course-specific strategies have also been implemented, like focusing on building community [19] and framing a course for elementary education majors within the context of pedagogy [54]. When students perceive course content as more relevant, they engage more and receive higher grades [55]. When a course is perceived as less relevant, students are less likely to retain the knowledge they learned [56].

Scholars have expressed concern that elementary education majors receive lower grades in general education courses than students in other majors [57,58]. These studies found that elementary education majors had lower grades in general education courses such as biology, English, and sociology. However, such evidence is from many years ago and the most recent research (still decades-old) indicates that elementary education majors did just as well in general education courses as students in other majors [59]. Additionally, these studies tend to rely on small samples.

Such differences, if they persist, could be because general education courses do not meet the needs of elementary education majors. Elementary education majors may not see general education courses as relevant for their personal interests or professional pursuits [18,19], though we are not aware of any research exploring perceptions of the relevance of general education coursework for preservice teachers. One of the leading drivers of perceived relevance is the perceived usefulness of the course [18]. Considering the low status of science in schools, it is likely that elementary education majors do not see the importance of developing science content knowledge in general education courses or see how the content taught will be involved in their future work [54]. There could also be other reasons that general education courses may not meet the needs of elementary education majors, including poor pedagogy [60], conflicts arising from elementary teachers' identities [61], or poor preparation from high school coursework [62]. Because of these concerns, many elementary education programs offer specialized science courses for future teachers as an alternative to general education

courses [16,63,64]. Such specialized courses may be a valuable solution, but they are challenging to implement and maintain.

While we focus on course grades in this study, we acknowledge that there are concerns about grades: they are highly idiosyncratic, based on the expectations and practices of the individual instructor [65–67]. Additionally, the meaning of a grade is unclear, possibly indicating the extent to which content was mastered or the extent to which students displayed dispositions such as conscientiousness and timeliness [68–70]. In fact, there is both evidence supporting [71] and refuting [27] the connection between grades in science courses and other measures of science content knowledge. Thus, both the reliability and validity of grades have been questioned.

Nonetheless, grades are an accepted indicator of academic success. Canfield, Kivisalu [72] found that grades in college courses were good indicators of meeting intended course outcomes. Similarly, Allensworth and Clark [73] found that high school grade-point average (GPA), the result of years of grades, was a strong predictor of success in college, stronger than ACT scores (a college-entrance exam). As such, in this study we explore grades in general education courses from the stance that grades are valuable, but contested, indicators of success.

## Methods

### Research context

This research took place at a large, private university in the western United States. As is common at universities in the United States, undergraduate students pursue a major, which entails a required series of courses specialized in an area of study. For example, a student majoring in elementary education completes courses related to child development, learning theories, and planning and enacting instruction in elementary classrooms. These are courses that students not majoring in elementary education would not complete.

In addition to the requirements for the major, all undergraduate students are expected to complete general education requirements. These require students to complete courses in several different categories. There are often multiple courses in each category. Sometimes students complete a course that is designed for a wide variety of majors, known as a general education course, while others complete a course that meets the general education requirements but is a part of their major program.

At this institution, students are expected to complete one course to meet the biological science requirement and another course to meet the physical science requirement. These are the only science content requirements for elementary education majors. Many choose to complete introductory biology and introductory physical science, which are general education courses, to meet these requirements. Students are also required to complete a general education course in American History. Most students at the university complete the same course because there are few other alternatives. Students must also complete courses in civilization, which includes a range of courses in the humanities, and many students select to complete course in western humanities. They must also complete a course in quantitative reasoning (many choose the general education course in statistics), and social sciences (many choose a human development course). Elementary education majors complete one science methods course as a part of the program, which brings emphasizes science pedagogy.

### Data source

Data for this study was drawn from the academic records housed in the university registrar's office. To gather these data, a request was placed with the academic records office. In this

request, we asked for data from students completing specific general education courses between 2009 – 2019 (data accessed on 6 July 2023). We chose these years because many of the students who completed these courses will be graduated at this point (and their major at graduation is one of the variables of interest). These years also avoid the emergency remote instruction that occurred during 2020 and 2021. As such, the results are not influenced by unusual instruction occurring during that time. Informed consent was not obtained from participants because data was completely de-identified, as approved by our institutional review board.

We collected data from all sections of the following courses during these years: introductory biology (Biology), introductory physical science (Physical Science), introductory statistics (Statistics), American history (History), western humanities (Humanities), and Human Development. Biology and Physical Science are among the most common science courses completed by elementary education majors at this university. Statistics is a mathematics-focused general education course completed by many elementary education majors. History and Humanities are non-science general education courses. Finally, Human Development is a social science course highly related to their future work as elementary teachers.

We gathered the final letter grades in each of these courses (GRADE). All grades are clustered by section (i.e., students who were taught by the same instructor at the same time) so that we can use SECTION as a variable in our analysis. We also received the major for each participant at the time of graduation. Because students may have changed their major, this may not have been their major at the time of completing the course. However, capturing major at graduation allows us to identify students who graduated with a license to teach elementary school and compare them with students who graduated with other majors.

The data set included 195860 grades from students in 230 different majors (See S1 for full data set). Because the data was archival and completely anonymous, obtaining consent from participants was not possible or necessary. The research team is not aware of the identity of participants. This procedure was approved by Brigham Young University's IRB (IRB2023-201).

## Data preparation

These data were prepared for analysis. First, GRADEs were converted from letter grades (e.g., A) to numerical values on a 4.0-scale, as depicted in Table 1. SECTION was used in our analysis without modification.

Table 1. Conversion from letter grade to values.

| Letter | Values |
|---|---|
| A | 4 |
| A- | 3.666 |
| B+ | 3.333 |
| B | 3 |
| B- | 2.666 |
| C+ | 2.33 |
| C | 2 |
| C- | 1.666 |
| D+ | 1.333 |
| D | 1 |
| D- | .666 |
| E/W/IE/V | 0 |

Students' majors were prepared for analysis by categorizing them into MAJOR TYPEs. These categories were created to cluster majors that include related coursework, ways of thinking, and professional pathways. Elementary Education (ELED) was included as its own MAJOR TYPE due to our interest in those students, and all majors other than ELED were clustered into broad fields, giving the following MAJOR TYPEs:

1. Arts and communications: Majors that focus on expressing human experience and ideas through media such as music, painting, video, and writing.

2. Business: Majors that focus on how organizations operate, make decisions, and manage resources.

3. Humanities: Majors in the humanities explore human society, culture, and thought, such as art history, languages, literature, and philosophy.

4. Social sciences: These majors focus on how people live together in groups and why they think or behave like they do.

5. Science, technology, engineering, and mathematics (STEM): These majors are about understanding and harnessing the natural and designed world.

6. Elementary Education (ELED): This category included early childhood education and elementary education majors. While there are important differences in the programs, both prepare students for teaching in elementary schools, and both have the same requirements for science content.

Secondary education majors were included with the majors of their disciplines (e.g., a student majoring in Physics Teaching was considered a STEM major) since the bulk of the coursework for secondary education programs are focused on disciplinary content and taught by disciplinary experts. Students (n = 2661, 1.4% of the total sample) who majored in "general studies" were excluded from these categories because the needed information was unspecified (i.e., general education majors pursue individualized programs of study). The number of students in each of these MAJOR TYPEs is captured in Table 2.

## Analysis

All analyses were conducted using IBM® SPSS Statistics (version 29.0.1.0). We chose to use linear mixed modeling to predict course grades, so we could use SECTION as a random effect to account for the nested nature of the data. By allowing for a random intercept for each section, we were able to compare elementary education majors to their peers in their same section. This eliminates confounding variables (e.g., course structure, pedagogical choices) that would impact student performance. We used a separate model for each general education course, with final course grade (GRADE) in that course as our outcome and major (MAJOR

**Table 2. Number of students in each MAJOR TYPE.**

| MAJOR TYPE | Number of Students | Percent |
|---|---|---|
| Arts and Communications | 21,752 | 11.3 |
| Business | 28,132 | 14.6 |
| Humanities | 17,668 | 9.1 |
| Social Sciences | 53,258 | 23.7 |
| STEM | 72,389 | 37.5 |
| ELED | 7,461 | 3.9 |

TYPE) as a possible fixed effect. Default settings for the MIXED command in SPSS were used unless otherwise noted. We followed the steps outlined by Theobald [74] for each course as follows.

First, we ran an empty model that included random intercepts for each section (i.e., GRADE ~ (1|SECTION)) in order to calculate the intraclass correlation coefficient for the random effect. For all courses, the ICC for SECTION was less than 0.1, and it was often less than 0.05 (Biology: 0.074; Physical Science: 0.051; Statistics: 0.038; Humanities: 0.051; Human Development: 0.037; History: 0.015). Based on this step alone, we were not confident the random effect was needed in our models.

To settle the question, we selected random effects by using the Akaike information criterion (AIC) to compare full models that did and did not include SECTION as a random effect (i.e., GRADE ~ MAJOR TYPE + (1|SECTION) compared to GRADE ~ MAJOR TYPE). Restricted maximum likelihood was used when fitting these models. For all courses, including SECTION improved the model (decreased the AIC), so we proceeded with including random intercepts by SECTION when modeling GRADEs in each course.

Next, we selected fixed effects by using the AIC to compare models with and without grouping by MAJOR TYPE (i.e., GRADE ~ MAJOR TYPE + (1|SECTION) compared to GRADE ~ (1|SECTION)). Maximum likelihood was used when fitting these models. For all courses, including MAJOR TYPE as a fixed effect improved the model (decreased the AIC), so we proceeded with including this in our final model for each course.

Finally, we reran the final model for each course (i.e., GRADE ~ MAJOR TYPE + (1|SECTION)) using restricted maximum likelihood to get accurate parameter estimates. Because MAJOR TYPE had multiple categories (see above), the 95% confidence intervals of parameter estimates were used to determine which broad fields were statistically distinguishable from each other with ELED as the reference MAJOR TYPE. We also calculated estimated marginal means of the grades for each MAJOR TYPE to be used in data visualizations. The results of this final step are included in the Results section below.

## Sample

To provide a clear understanding of our sample and facilitate comparisons with other populations, we also gathered ACT score and high school GPA data for the students in our dataset. We used one-way ANOVAs to compare by MAJOR TYPE. For ACT score, there was a significant effect for MAJOR TYPE, $F(5, 1183215) = 730.20$, $p < 0.001$), and post-hoc Tukey comparisons between ELED and the other majors showed that the mean score for ELED majors (M = 26.56, SD = 3.26, N = 7296) was significantly lower than the mean score for Business (M = 27.47, SD = 3.49, N = 26631), Humanities (M = 27.63, SD = 3.48, N = 16669), and STEM majors (M = 27.66, SD = 3.71, N = 69238). For high school GPA, there was also a significant effect for MAJOR TYPE, $F(5, 170121) = 340.01$, $p < 0.001$, and post-hoc Tukey comparisons between ELED and the other majors showed that the mean GPA for ELED majors (M = 3.81, SD = 0.21, N = 6903) was significantly higher than all other major types (Arts and Communications: M = 3.72, SD = 0.28, N = 19297; Business: M = 3.77, SD = 0.27, N = 25286; Humanities: M = 3.72, SD = 0.28, N = 15508; Social Sciences: M = 3.72, SD = 0.28, N = 38978; STEM: M = 3.77, SD = 0.26, N = 64155).

Because ELED majors differed from their peers in significant ways (based on ACT and high school GPA), we could have used these variables as covariates to account for the impacts of these differences on final grades before comparing by major type. However, we were not interested in whether general education courses served ELED majors after accounting for incoming differences. Rather, we wanted to investigate how ELED majors compared to other students regardless of background because we are interested in what knowledge ELED majors

are bringing with them to teaching. Thus, we do not use ACT and high school GPA as covariates in our study.

### Limitations

To capture such a large set of data, it was necessary that we accept some limitations. The primary limitation is that we know that grades are not necessarily an indicator of learning [75,76]. Grades capture a wide span of indicators, often including attendance, participation, and timeliness [77]. However, it is likely impossible to get more detailed data on student learning for such a large sample. Therefore, we confine our claims to the grades students received in these courses, carefully avoiding making claims about student learning.

Another important limitation is that we have no data about the specific sections of the courses or specific students. For example, there are differences in how sections were taught over the years. Furthermore, there may also be characteristics of individual students that influenced their experience in these courses (e.g., perception of relevance). With a large, retrospective data set we were unable to gather such information. While we will speculate on potential causes, we do not have evidence of these causes.

## Results

We begin with general descriptive statistics to provide an overview of the data set. We then present the results of the statistical models.

### Sample characteristics

As we report the characteristics of participants in this sample, we should note that we do not have information about how many individuals are represented by this data set. Our data is associated with a grade in one of the six courses: an individual may have taken a course multiple times and individuals likely completed more than one of the six courses.

Of the six courses in this sample, the largest portion of grades came from Biology, History, and Statistics, with each of these accounting to approximately a quarter of the grades (see Table 3). The remaining quarter of grades were drawn from the remaining three courses. Importantly, this difference is a result of differences in actual enrollment rather than sampling since all grades from these courses were collected.

Mean grades for each course are presented in Table 3. There is a statistically significant difference among the mean grades in these courses, $F(5, 195859) = 1142$, $p < .001$, $h^2 = .028$. The lowest mean grades were in History and Statistics, whereas the highest mean grades were in Biology, Humanities, and Physical Science.

**Table 3. Students and GRADES in each course.**

| | | BIO | PHYS | STAT | HUM | HDEV | HIST |
|---|---|---|---|---|---|---|---|
| Number of Grades | Number | 40,113 | 34,439 | 46,705 | 12,276 | 14,732 | 47,595 |
| | Percentage | 20.5% | 17.6% | 23.8% | 6.3% | 7.5% | 24.3% |
| GRADE | Mean | 3.16 | 3.17 | 2.80 | 3.16 | 3.03 | 2.77 |
| | Median | 3.33 | 3.66 | 3.33 | 3.66 | 3.33 | 3.00 |
| | SD | 1.01 | 0.98 | 1.24 | 1.02 | 1.02 | 1.04 |

Note: Abbreviations in the table are as follows: Biology (BIO), Physical Science (PHYS), Statistics (STAT), Humanities (HUM), Human Development (HDEV), and History (HIST).

## Model results

As outlined in the Methods, we used linear mixed models to model final course grades in each general education course separately. As indicated in the Methods, the final model for each course was GRADES ~ MAJOR TYPE + (1|SECTION) after selection of random and fixed effects. Fig 2 shows the estimated marginal means for each major in each course, and Table 4 shows parameter estimates for fixed effects in the final models. We compared ELED majors to their peers from other disciplines based on overlap (or lack of overlap) between parameter estimates' 95% confidence intervals.

As shown in Fig 2 and Table 4, ELED majors earned similar grades in Biology as STEM majors, but lower grades than business majors and higher grades than everyone else. In Physical Science, ELED majors earned lower grades than business and STEM majors, but they earned higher grades than everyone else. In Statistics, ELED majors earned lower grades than business and STEM majors, similar grades as humanities majors, and higher grades than arts & communications and other social science majors. In Humanities, ELED majors earned lower grades than business, humanities, and STEM majors, but they earned higher grades than arts & communications and other social science majors. In Human Development, ELED majors earned lower grades than business and STEM majors, but they earn higher grades than everyone else. Finally, in History, ELED majors earned lower grades than business, humanities, and STEM majors, but they earned higher grades than arts & communications and other social science majors.

## Discussion

This is the first study in many years, of which we are aware, to explore the impact of general education courses on preservice teachers [57–59]. In addition to being more recent, this study draws on a larger sample of students over a longer time span than previous research.

Overall, these results indicate that elementary education majors are receiving grades in general education courses similar to students in most other majors, as found in [59]. This suggests that these general education courses are not differentially disadvantaging elementary education majors [61]. Furthermore, elementary education majors are doing quite well in general education courses, averaging in the range of an A or A- grade in most courses. These results assuage some concerns from past scholars about elementary education majors not

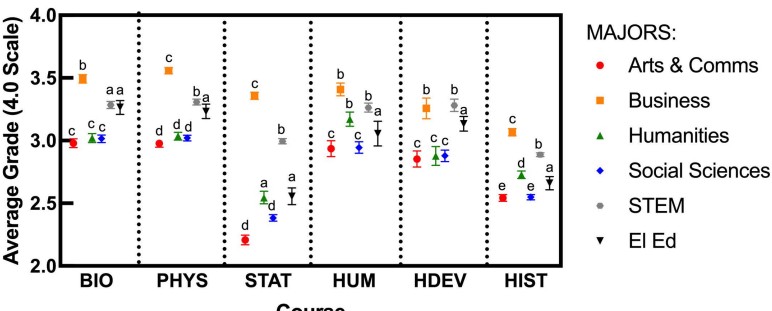

**Fig 2. Final course grades of elementary education majors compared to other majors in each general education course.** The average course grades on the x axis were calculated as estimated marginal means following linear mixed modeling for each course. Major type was included as a fixed effect, and section as a random effect (GRADE ~ MAJOR TYPE + (1|SECTION)). Error bars are 95% confidence intervals. The small letters indicate statistical equivalence or difference based on 95% confidence interval overlap of model parameter estimates. As each course was modeled separately, no conclusions can be drawn regarding differences across the dashed lines.

**Table 4. Fixed effects parameter estimates for each course's final linear mixed model: GRADE ~ MAJOR TYPE + (1|SECTION).**

| Course | Fixed Effect | B | SE | t | p | 95%CI | n |
|---|---|---|---|---|---|---|---|
| Biology | Intercept | 3.265 | 0.028 | 115.949 | <0.001 | [3.210, 3.320] | 4830 |
| | Arts & Comms | -0.286 | 0.029 | -9.826 | <0.001 | [-0.343, -0.229] | 6097 |
| | Business | 0.228 | 0.028 | 7.994 | <0.001 | [0.172, 0.283] | 4479 |
| | Humanities | -0.245 | 0.029 | -8.354 | <0.001 | [-0.303, -0.188] | 9399 |
| | Social Sciences | -0.250 | 0.027 | -9.100 | <0.001 | [-0.304, -0.196] | 13473 |
| | STEM | 0.019 | 0.027 | 0.701 | 0.483 | [-0.034, 0.072] | 1399 |
| | Elementary Ed[a] | | | | | | |
| Physical Science | Intercept | 3.233 | 0.029 | 110.935 | <0.001 | [3.176, 3.290] | 4898 |
| | Arts & Comms | -0.256 | 0.031 | -8.147 | <0.001 | [-0.318, -0.195] | 6550 |
| | Business | 0.324 | 0.031 | 10.545 | <0.001 | [0.264, 0.385] | 4075 |
| | Humanities | -0.198 | 0.032 | -6.163 | <0.001 | [-0.261, -0.135] | 9244 |
| | Social Sciences | -0.212 | 0.030 | -7.041 | <0.001 | [-0.271, -0.153] | 8118 |
| | STEM | 0.074 | 0.030 | 2.455 | 0.014 | [0.015, 0.134] | 1084 |
| | Elementary Ed[a] | | | | | | |
| Statistics | Intercept | 2.556 | 0.034 | 74.722 | <0.001 | [2.489, 2.623] | 4035 |
| | Arts & Comms | -0.348 | 0.038 | -9.082 | <0.001 | [-0.423, -0.273] | 8008 |
| | Business | 0.802 | 0.036 | 22.212 | <0.001 | [0.731, 0.873] | 2192 |
| | Humanities | -0.011 | 0.042 | -0.260 | 0.795 | [-0.093, 0.071] | 9791 |
| | Social Sciences | -0.173 | 0.036 | -4.851 | <0.001 | [-0.243, -0.103] | 20654 |
| | STEM | 0.438 | 0.035 | 12.663 | <0.001 | [0.371, 0.506] | 1204 |
| | Elementary Ed[a] | | | | | | |
| Humanities | Intercept | 3.056 | 0.050 | 60.598 | <0.001 | [2.958, 3.155] | 1095 |
| | Arts & Comms | -0.120 | 0.057 | -2.098 | 0.036 | [-0.233, -0.008] | 1806 |
| | Business | 0.352 | 0.054 | 6.471 | <0.001 | [0.245, 0.458] | 1385 |
| | Humanities | 0.114 | 0.056 | 2.050 | 0.040 | [0.005, 0.223] | 2464 |
| | Social Sciences | -0.112 | 0.053 | -2.113 | 0.035 | [-0.216, -0.008] | 4899 |
| | STEM | 0.207 | 0.051 | 4.045 | <0.001 | [0.107, 0.307] | 404 |
| | Elementary Ed[a] | | | | | | |
| Human Development | Intercept | 3.134 | 0.030 | 105.223 | <0.001 | [3.076, 3.193] | 1413 |
| | Arts & Comms | -0.281 | 0.035 | -8.059 | <0.001 | [-0.349, -0.213] | 699 |
| | Business | 0.123 | 0.044 | 2.810 | 0.005 | [0.037, 0.209] | 904 |
| | Humanities | -0.256 | 0.040 | -6.397 | <0.001 | [-0.335, -0.178] | 5506 |
| | Social Sciences | -0.255 | 0.027 | -9.624 | <0.001 | [-0.307, -0.203] | 4059 |
| | STEM | 0.147 | 0.028 | 5.331 | <0.001 | [0.093, 0.201] | 1889 |
| | Elementary Ed[a] | | | | | | |
| History | Intercept | 2.661 | 0.027 | 99.845 | <0.001 | [2.608, 2.713] | 5481 |
| | Arts & Comms | -0.119 | 0.030 | -3.971 | <0.001 | [-0.177, -0.060] | 4972 |
| | Business | 0.406 | 0.030 | 13.463 | <0.001 | [0.347, 0.466] | 4633 |
| | Humanities | -0.067 | 0.030 | 2.199 | 0.028 | [0.007, 0.127] | 9393 |
| | Social Sciences | -0.112 | 0.029 | -3.929 | <0.001 | [-0.168, -0.056] | 21186 |
| | STEM | 0.226 | 0.027 | 8.256 | <0.001 | [0.173, 0.280] | 1481 |
| | Elementary Ed[a] | | | | | | |

[a]For MAJOR, Elementary Education was the reference category.

being served by general education courses [57,58]. An implication is that these general education courses seem to be serving elementary education majors, at least as indicated by grades, as well as students with other majors. Future research should explore other indicators of how well general education courses are preparing future elementary teachers, such as alignment with topics taught in elementary grades [17] or other measures of content knowledge [78].

It seems that elementary education majors do not find these general education courses irrelevant, at least not to the point of disengaging with general education courses more than students in other majors. This mitigates some concerns from past research [18,19]. Admittedly, this relevance could come from a low-level sense that the course is required for

graduation and, even then, could simply be at the level of making the grade rather than deeply understanding [79]. Future research should more directly explore elementary education majors' perception of relevance in general education courses.

Of primary interest is elementary education majors' performance in science courses. It is notable that elementary education majors received grades comparable to STEM majors in Biology, while scoring worse than STEM majors in Physical Science. This difference suggests elementary education majors would benefit more from a specialized course in physical science than in biology. As such, elementary education programs seeking to provide a specialized science course [16,63,64] may want to prioritize a course in physical science.

This may result from, or contribute to, elementary education majors' preferential confidence in life sciences over physical sciences [33–35]. While not addressed with our data, it is possible that the differences we observed in Biology versus Physical Science could reflect broader trends in gender differences. Because most of the elementary education majors in our sample were surely female [36–38], it is reasonable to expect these data to reflect patterns from previous research. We do, in fact, observe this; past research has shown gender differences in STEM courses, with female students often receiving lower grades than male in STEM courses [43–46]. However, even with receiving lower grades than STEM majors in Physical Science, elementary education majors scored above the average grades for students majoring in arts and communications, humanities, and social sciences, all of which likely have higher proportions of male students than elementary education majors. Thus, more research would be needed to clarify the relationships between elementary education major, gender, and performance in different types of STEM courses. Finally, it should be noted that elementary education majors only scored 0.074 grade points lower than STEM majors on average in Physical Science, which equates to less than a fourth of the difference between an A and A-. Thus, while statistically significant in our large dataset, it is not practically meaningful.

It is important to note two other contextual factors that may influence these differences among majors. The first of these contextual factors is that some STEM majors are unlikely to complete Biology or Physical Science courses. Rather than completing the general education version of these courses, they are expected to complete the introductory course designed specifically for some majors (e.g., biology majors complete a majors-only introductory course instead of the general education Biology course). As such, STEM majors who completed Biology or Physical Science either were not majoring in the relevant STEM major or are students who changed majors after completing the general education course. The second contextual factor has to do with the admissions requirements for business majors. At this institution, business majors have some of the most stringent program admissions requirements. Many students who apply for a business major will have completed these general education courses. Those who did not receive high grades in general education courses are unlikely to be admitted to the program.

In addition to exploring elementary education majors' perceptions of the relevance of general education courses, future research should examine other indicators of elementary teacher content learning in general education courses. While this study relies on grades, we acknowledge that other indicators of knowledge would provide additional evidence of the impact general education courses [22]. Further developing this knowledge would support efforts to better prepare elementary teachers with the science content knowledge they need for teaching children.

## Supporting information

**S1 File. Data set.** Full data set analyzed for this study.
(XLSX)

## Author contributions

**Conceptualization:** Ryan S. Nixon, Elizabeth Gibbons Bailey.

**Data curation:** Ryan S. Nixon, Elizabeth Gibbons Bailey.

**Formal analysis:** Elizabeth Gibbons Bailey.

**Methodology:** Ryan S. Nixon, Elizabeth Gibbons Bailey.

**Project administration:** Ryan S. Nixon.

**Visualization:** Elizabeth Gibbons Bailey.

**Writing – original draft:** Ryan S. Nixon, Elizabeth Gibbons Bailey.

**Writing – review & editing:** Ryan S. Nixon, Elizabeth Gibbons Bailey.

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
