## [Decision Letter · Decision Letter 0]

15 Dec 2024

PONE-D-24-44037Comparing Academic Performance of Elementary Education Majors in General Education Science CoursesPLOS ONE

Dear Dr. Nixon,

Thank you for submitting your manuscript to PLOS ONE. After careful consideration, we feel that it has merit but does not fully meet PLOS ONE’s publication criteria as it currently stands. Therefore, we invite you to submit a revised version of the manuscript that addresses the points raised during the review process.

We look forward to receiving your revised manuscript.

Kind regards,

Miguel Ángel Queiruga-Dios, Ph.D.

Academic Editor

PLOS ONE

Journal Requirements: When submitting your revision, we need you to address these additional requirements. 1. Please ensure that your manuscript meets PLOS ONE's style requirements, including those for file naming. The PLOS ONE style templates can be found at https://journals.plos.org/plosone/s/file?id=wjVg/PLOSOne_formatting_sample_main_body.pdf and https://journals.plos.org/plosone/s/file?id=ba62/PLOSOne_formatting_sample_title_authors_affiliations.pdf 2. Please include your full ethics statement in the ‘Methods’ section of your manuscript file. In your statement, please include the full name of the IRB or ethics committee who approved or waived your study, as well as whether or not you obtained informed written or verbal consent. If consent was waived for your study, please include this information in your statement as well. 3. Please include captions for your Supporting Information files at the end of your manuscript, and update any in-text citations to match accordingly. Please see our Supporting Information guidelines for more information: http://journals.plos.org/plosone/s/supporting-information. 4. Please review your reference list to ensure that it is complete and correct. If you have cited papers that have been retracted, please include the rationale for doing so in the manuscript text, or remove these references and replace them with relevant current references. Any changes to the reference list should be mentioned in the rebuttal letter that accompanies your revised manuscript. If you need to cite a retracted article, indicate the article’s retracted status in the References list and also include a citation and full reference for the retraction notice.

**Additional Editor Comments:**

Dear authors,

Thank you very much for sending your manuscript and for your patience while waiting for the review.

Below are the reviewers' comments.

Please try to respond to the reviewers' requests.

Reviewers' comments:

Reviewer's Responses to Questions

**Comments to the Author**

1. Is the manuscript technically sound, and do the data support the conclusions?

Reviewer #1: Partly

Reviewer #2: Partly

Reviewer #3: Yes

Reviewer #4: Yes

Reviewer #5: Yes

2. Has the statistical analysis been performed appropriately and rigorously? 

Reviewer #1: Yes

Reviewer #2: Yes

Reviewer #3: Yes

Reviewer #4: I Don't Know

Reviewer #5: Yes

3. Have the authors made all data underlying the findings in their manuscript fully available?

Reviewer #1: Yes

Reviewer #2: Yes

Reviewer #3: Yes

Reviewer #4: No

Reviewer #5: Yes

4. Is the manuscript presented in an intelligible fashion and written in standard English?

Reviewer #1: Yes

Reviewer #2: Yes

Reviewer #3: Yes

Reviewer #4: Yes

Reviewer #5: Yes

5. Review Comments to the Author

Reviewer #1: Thank you for the opportunity to review your research on this intricate and thought-provoking topic. Your dedication to addressing such a challenging area is commendable. While I found the methodology and analysis to be sound, I have several suggestions regarding the introduction, literature review (LR), and discussion sections that could enhance the clarity and depth of the study.

One key issue is the comparability of the student groups analyzed. It is unclear whether these groups share similar academic and cognitive characteristics at the point of university admission. Are they assessed and admitted using comparable criteria, and do they begin with equivalent levels of knowledge and skills? Establishing that the groups are indeed similar is essential, as this underpins the rationale for expecting comparable academic performance. Providing additional evidence on their profiles would strengthen the foundation of the comparisons made.

In the rationale section, the distinction between elementary education majors and other groups could be more clearly articulated. Specifically, what aspects of their educational backgrounds set elementary education majors apart from students in other disciplines? Clarifying this distinction would help readers better understand the relevance and purpose of the comparisons in your study.

Although the LR section presents a compelling argument, it might be worth considering that institutions offering teacher education programs may not prioritize science competency as a standalone outcome. These programs may instead focus on integrating content knowledge with pedagogical training. However, the study does not provide sufficient detail about the curricular focus of these programs. Including such information would help contextualize your findings and provide a clearer picture of the institutional goals.

Additionally, the concern raised about the relevance of general education courses may not be unique to elementary education majors. Students from other disciplines could also view these courses as less directly applicable, as they primarily focus on content knowledge rather than its integration with discipline-specific pedagogical methods. Expanding on this broader perspective would add depth to your argument.

While the results indicate similar performance levels between elementary education majors and other students in general education courses, they may not fully capture important aspects of teacher readiness. Academic performance, as measured by grades, might overlook critical elements such as pedagogical skills, subject matter expertise, and practical teaching experience.

The lower performance of elementary education majors in Physical Science compared to STEM majors is particularly noteworthy. This could reflect differences in the design and focus of teacher education curricula or even individual personality traits influencing career choices. A discussion of these potential factors would provide a more nuanced interpretation of your findings.

Furthermore, achieving similar content knowledge does not necessarily translate into equivalent pedagogical potential. High grades might reflect a focus on maintaining a strong GPA rather than genuine engagement with the material's relevance to teaching. Addressing this limitation and including counterarguments would help present a balanced discussion.

The study would benefit from a more detailed exploration of its implications. How do these findings inform the preparation of future educators or the design of teacher education programs? Are there recommendations for making general education courses more relevant to teacher preparation? Highlighting these aspects would enhance the study’s contribution to the field.

In conclusion, addressing these points would improve the clarity and comprehensiveness of the introduction, literature review, and discussion sections, ultimately making the study more impactful. I look forward to seeing how these suggestions are incorporated in the revised version.

Reviewer #2: Abstract and Introduction: Include specific implications for educational practice and policy in the abstract and introduction.

Methods: Provide a rationale for relying on grades and consider acknowledging potential confounders more explicitly. Limitations related to potential sampling biases (e.g., students changing majors) are only briefly mentioned and could be elaborated on.

Results: Explore potential mechanisms driving differences between groups and address variations in course grading rigor more thoroughly. The results section could benefit from more detailed exploration of why elementary education majors perform similarly to or differently from other groups in specific courses, beyond referencing existing literature.

Discussion: Expand on practical recommendations for improving general education courses and propose concrete strategies for enhancing science education for preservice teachers.

Conclusions: Highlight broader implications for teacher preparation programs and elementary education.

References: Some references to "Author" placeholders are incomplete.

Reviewer #3: The paper addresses a common misconception that elementary education students do not perform as well as other students on campus. This is presented as a comparison of grades across general education courses with other categories of majors. The authors directly note that grade attainment is not the same as learning the content. The study is well designed, and the data analyses are reasonable for the claims they are making. The writing is clear and error free. I recommend accepting the paper.

Reviewer #4: Abstract:

- The research goal should be rephrased, make it more clear for the reader.

- The work significance for the research field should be detailed.

I could not revise the figures: they had not been provided to me.

Suggestions of rephrasing:

- “As noted by Rice (2005, p. 1078), there is an assumption that preservice elementary teachers “entering teacher education [programs] have adequate science subject matter knowledge" from their prior coursework.

- Therefore, our research question is: How do the grades of elementary education

majors compare to the grades of other students in general education courses?

- “Canfield and colleagues (2015) found”. Canfield et al. (2015)

- “found that high school GPA, the result of years of grades, was a strong predictor of success in college, stronger than ACT scores”. When you use, for the first time, one acronym, please explain it. Another example: “ELED was included as its own…”

- Table 1 is more useful if it appears next to the place where it is cited. The same comment for the other tables and images.

- Please set a criteria and always follow it: use , or . (n = 2,661, 1.4% of the total sample). Is it 2661, or 2,661?

Phrases that I did not understand:

- a faculty member who teaches these general education courses becomes a “de facto…teacher educator” (Grossman et al., 1989, p. 25).

- “Admittedly, this relevance could come from a low-level sense that the course is required for graduation and, even then, could simply be at the level of making the grade rather than deeply understanding”. It is difficult to find results that clearly support this statement.

The paper presents a very interesting literature review. However, the 63 works cited have an average age of 2009: it is more than 15 years old. Therefore, we suggest an effort, from the authors to update this literature review.

This great literature review is not, afterwords, used in the discussion section. Of 63 works cited, only 9 are used to confront the author’s results, in the discussion section. That option is very questionable. It is, indeed, unintelligible: a great literature review which is not used to discuss the results.

It is missing the bibliographic reference for “Evagorou et al., 2022”.

Reviewer #5: The study addresses an important topic by exploring the academic performance of elementary education majors in general education science courses.However, more context about how the findings inform teacher preparation programs would strengthen the paper. Consider rephrasing the conclusion to emphasize the implications for teacher preparation programs. Adding specific examples of the content mismatch between general education science courses and elementary teaching requirements could enhance the argument. The results are well-organized. Figures and tables are clear, but including more visualizations, such as boxplots or scatter plots, might help readers grasp the variability in grades across majors and sections. Confidence intervals and significance levels are reported effectively. Adding effect sizes could further contextualize the practical significance of findings. The discussion highlights the relevance of grades as an indicator of success but could benefit from deeper exploration of how perceptions of relevance impact performance. The authors acknowledge the limitations of using grades as a proxy for learning. Minor grammatical issues and awkward phrasing exist (e.g., "While we focus on course grades in this study" in the introduction). A thorough proofreading is recommended. While the manuscript presents a valuable and well-conducted study, several areas need refinement, particularly in connecting findings to implications for teacher preparation and enhancing clarity in data presentation. Addressing these issues will strengthen the manuscript and its contribution to the field.

6. PLOS authors have the option to publish the peer review history of their article (what does this mean? ). If published, this will include your full peer review and any attached files.

**Do you want your identity to be public for this peer review?** For information about this choice, including consent withdrawal, please see our Privacy Policy .

Reviewer #1: No

Reviewer #2: No

Reviewer #3: No

Reviewer #4: No

Reviewer #5: No

---

## [Author Response · Author response to Decision Letter 1]

28 Jan 2025

The fully formatted was uploaded as a cover letter. Please see that for clearer formatting.

Dear editors,

Thank you for the opportunity to respond to this feedback.

We have addressed the following requirements:

1. We have modified the files to follow the specified conventions.

2. We have added an ethics statement to the Methods section.

3. We have added a caption for Supporting Information.

4. We have reviewed the reference list. We have not cited any retracted manuscripts.

Below we respond to each of the comments from the reviewers.

Reviewer Comments Author Response

Reviewer 1

One key issue is the comparability of the student groups analyzed. It is unclear whether these groups share similar academic and cognitive characteristics at the point of university admission. Are they assessed and admitted using comparable criteria, and do they begin with equivalent levels of knowledge and skills? Establishing that the groups are indeed similar is essential, as this underpins the rationale for expecting comparable academic performance. Providing additional evidence on their profiles would strengthen the foundation of the comparisons made. We have added a section to the Methods that compares ELED majors to other majors in terms of ACT and high school GPA. This allows readers to compare our sample to their own institution, and it gives an idea of how ELED majors’ preparation compares to their peers in our population. However, establishing that the groups are equivalent is not critical (or even desirable) for our research question. We are interested in how ELED majors leave their general education courses, regardless of preparation, since this is the knowledge with which they enter their teaching jobs. We don’t want to answer the question of whether ELED majors are comparable to STEM majors all things considered, but rather whether they are comparable to STEM majors, period. We have added a better explanation of this to that same new section in the Methods.

In the rationale section, the distinction between elementary education majors and other groups could be more clearly articulated. Specifically, what aspects of their educational backgrounds set elementary education majors apart from students in other disciplines? Clarifying this distinction would help readers better understand the relevance and purpose of the comparisons in your study. We have added some information comparing academic preparation of ELED majors vs other majors (see previous comment). We have also more explicitly indicated that ELED majors tend to be female and the implications for that in college science courses.

Although the LR section presents a compelling argument, it might be worth considering that institutions offering teacher education programs may not prioritize science competency as a standalone outcome. These programs may instead focus on integrating content knowledge with pedagogical training. However, the study does not provide sufficient detail about the curricular focus of these programs. Including such information would help contextualize your findings and provide a clearer picture of the institutional goals. We have added some about the program in the Research Context section. Here we mention the science methods course that brings together content and pedagogy.

Additionally, the concern raised about the relevance of general education courses may not be unique to elementary education majors. Students from other disciplines could also view these courses as less directly applicable, as they primarily focus on content knowledge rather than its integration with discipline-specific pedagogical methods. Expanding on this broader perspective would add depth to your argument. The literature related to relevance of general education courses is not specific to elementary education majors. Instead, this research is conducted with a broad, general audience. We have clarified this.

While the results indicate similar performance levels between elementary education majors and other students in general education courses, they may not fully capture important aspects of teacher readiness. Academic performance, as measured by grades, might overlook critical elements such as pedagogical skills, subject matter expertise, and practical teaching experience. We have added an acknowledgment of this in the first paragraph.

The lower performance of elementary education majors in Physical Science compared to STEM majors is particularly noteworthy. This could reflect differences in the design and focus of teacher education curricula or even individual personality traits influencing career choices. A discussion of these potential factors would provide a more nuanced interpretation of your findings. Yes, that was an interesting result. We have added a paragraph to the Discussion that talks about the potential confounding factor of gender (as most elementary education majors are female). We connected this to literature on gender differences in the biological vs physical sciences, and we ultimately called for more research since our dataset did not include sex or gender.

Furthermore, achieving similar content knowledge does not necessarily translate into equivalent pedagogical potential. High grades might reflect a focus on maintaining a strong GPA rather than genuine engagement with the material's relevance to teaching. Addressing this limitation and including counterarguments would help present a balanced discussion. We have added an acknowledgment of this in the first paragraph.

The study would benefit from a more detailed exploration of its implications. How do these findings inform the preparation of future educators or the design of teacher education programs? Are there recommendations for making general education courses more relevant to teacher preparation? Highlighting these aspects would enhance the study’s contribution to the field. Thank you for pointing this out. We have added implications, for research and practice, to the Discussion section. We have been careful to not overstep our data by making implications for pedagogy in general education courses because we do not have any evidence related to the instruction that occurred in these courses.

Reviewer 2

Methods: Provide a rationale for relying on grades and consider acknowledging potential confounders more explicitly. This is an important limitation to present. We discuss this in the Limitations section of the Methods.

Limitations related to potential sampling biases (e.g., students changing majors) are only briefly mentioned and could be elaborated on. We added a more explicit statement about the limitation of choosing major at graduation as our grouping variable.

Results: Explore potential mechanisms driving differences between groups and address variations in course grading rigor more thoroughly. We added a sentence to the first paragraph of the Analysis section of the Methods to more explicitly explain how the random intercept included in the models accounts for these differences between groups (since we did not have data about course grading, etc.).

The results section could benefit from more detailed exploration of why elementary education majors perform similarly to or differently from other groups in specific courses, beyond referencing existing literature. Because this study draws on an extensive, archival data set, we were unable to draw on more detailed data about why differences exist. We do explore some possibility, but this data was not available for this study. As a result, we rely on existing literature and point to future research.

Discussion: Expand on practical recommendations for improving general education courses and propose concrete strategies for enhancing science education for preservice teachers. Please see the response to Reviewer #1’s comment about implications.

Conclusions: Highlight broader implications for teacher preparation programs and elementary education. Please see the response to Reviewer #1’s comment about implications.

References: Some references to "Author" placeholders are incomplete. We have included the full citations.

Reviewer 4

The research goal should be rephrased, make it more clear for the reader. This feedback is not sufficiently specific to make changes.

The work significance for the research field should be detailed. This feedback is not sufficiently specific to make changes.

Suggestions of rephrasing:

- “As noted by Rice (2005, p. 1078), there is an assumption that preservice elementary teachers “entering teacher education [programs] have adequate science subject matter knowledge" from their prior coursework.

- Therefore, our research question is: How do the grades of elementary education

majors compare to the grades of other students in general education courses?

- “Canfield and colleagues (2015) found”. Canfield et al. (2015)

- “found that high school GPA, the result of years of grades, was a strong predictor of success in college, stronger than ACT scores”. When you use, for the first time, one acronym, please explain it. Another example: “ELED was included as its own…”

- Table 1 is more useful if it appears next to the place where it is cited. The same comment for the other tables and images.

- Please set a criteria and always follow it: use , or . (n = 2,661, 1.4% of the total sample). Is it 2661, or 2,661?

-Made this change.

-Made this change.

-Made this change.

-Made these changes.

-In accordance with the Submission Guidelines, “Do not include figures in the main manuscript file. Each figure must be prepared and submitted as an individual file”

-We have modified this to be consistent.

Phrases that I did not understand:

- a faculty member who teaches these general education courses becomes a “de facto…teacher educator” (Grossman et al., 1989, p. 25).

- “Admittedly, this relevance could come from a low-level sense that the course is required for graduation and, even then, could simply be at the level of making the grade rather than deeply understanding”. It is difficult to find results that clearly support this statement.

-Upon revisiting this, we noted that the sentence was unclear and was not needed. We have removed it.

-We do not have results that support this statement. Because of this, we have signaled our tentativeness by using the word “could” twice in that sentence. We have also followed this sentence with a sentence about future research exploring this possibility.

The paper presents a very interesting literature review. However, the 63 works cited have an average age of 2009: it is more than 15 years old. Therefore, we suggest an effort, from the authors to update this literature review. Thank you for noting this. We have revisited the literature and added citations to some recent research.

This great literature review is not, afterwords, used in the discussion section. Of 63 works cited, only 9 are used to confront the author’s results, in the discussion section. That option is very questionable. It is, indeed, unintelligible: a great literature review which is not used to discuss the results. Thank you for pointing this out. It’s clear that we did not incorporate the literature sufficiently. We have made modifications and now cite 22 pieces in our Discussion.

It is missing the bibliographic reference for “Evagorou et al., 2022”. Thanks for noting this. This has been corrected.

Reviewer 5

However, more context about how the findings inform teacher preparation programs would strengthen the paper. Please see the response to Reviewer #1’s comment about implications.

Consider rephrasing the conclusion to emphasize the implications for teacher preparation programs. Please see the response to Reviewer #1’s comment about implications.

Adding specific examples of the content mismatch between general education science courses and elementary teaching requirements could enhance the argument. Examples were added to the Introduction.

Figures and tables are clear, but including more visualizations, such as boxplots or scatter plots, might help readers grasp the variability in grades across majors and sections. We agree that showing more information about the variability in data would be desirable. The reason we chose to only show estimated marginal means (with error) is because of the nested nature of our data. If we were to show box plots comparing the different majors, it would visually imply that all samples were independent of each other which is untrue. For example, a box plot could hypothetically show elementary education majors lower than other majors simply because a cluster of elementary education majors took a specific section together that was more difficult than other sections. Thus, the way we chose to visualize the data is the most accurate way to show differences between majors (comparing them only to their in-section peers after the model is calculated).

Adding effect sizes could further contextualize the practical significance of findings. This was a great point. We have added to the discussion section to point out how small these significant differences are when it comes to practical grade differences.

The discussion highlights the relevance of grades as an indicator of success but could benefit from deeper exploration of how perceptions of relevance impact performance. We agree that this would be beneficial. Unfortunately, we do not have data about participants’ perceptions of relevance.

Minor grammatical issues and awkward phrasing exist (e.g., "While we focus on course grades in this study" in the introduction). This feedback is not sufficiently specific to make changes.

A thorough proofreading is recommended. Thank you. We have done this.

---

## [Editor Report · Decision Letter 1]

14 Feb 2025

Comparing Academic Performance of Elementary Education Majors in General Education Science Courses

PONE-D-24-44037R1

Dear Dr. Nixon,

We’re pleased to inform you that your manuscript has been judged scientifically suitable for publication and will be formally accepted for publication once it meets all outstanding technical requirements.

Kind regards,

Miguel Ángel Queiruga-Dios, Ph.D.

Academic Editor

PLOS ONE

Additional Editor Comments (optional):

Dear author,

Thank you very much for the response to the reviewers and for sending this manuscript, which enriches the scientific literature.

Kind Regards,
---

## [Editor Report · Acceptance letter]

PONE-D-24-44037R1

PLOS ONE

Dear Dr. Nixon,

I'm pleased to inform you that your manuscript has been deemed suitable for publication in PLOS ONE. Congratulations! Your manuscript is now being handed over to our production team.

Kind regards,

on behalf of

Dr. Miguel Ángel Queiruga-Dios

Academic Editor

PLOS ONE